What do giant titanosaur dinosaurs and modern Australasian megapodes have in common?

Hechenleitner E. Martín 1 emhechenleitner@gmail.com
Grellet-Tinner Gerald 1 2
Fiorelli Lucas E. 1
1 Department of Geosciences, Centro Regional de Investigaciones Científicas y Transferencia Tecnológica de La Rioja (CRILAR-CONICET) , La Rioja , Argentina
2 Orcas Island Historical Museum , Eastsound, Washington , United States
Young Mark
Electronic publication date: 2015 Oct 20
Publication date: 2015
Volume: 3
Electronic Location ID: e1341
Received 2015 Jun 28; Accepted 2015 Sep 30
Copyright: © 2015 Hechenleitner et al.
Copyright year: 2015
Copyright holder: Hechenleitner et al.
License: This is an open access article distributed under the terms of the Creative Commons Attribution License, which permits unrestricted use, distribution, reproduction and adaptation in any medium and for any purpose provided that it is properly attributed. For attribution, the original author(s), title, publication source (PeerJ) and either DOI or URL of the article must be cited.
License URL: https://creativecommons.org/licenses/by/4.0/

Keywords: Nesting environment, Nesting site, Incubation with environmental heat, Labile nesting behavior, Egg physiology, Titanosaur

Funding: Secretaría de Gobierno of La Rioja CONICET Jurassic Foundation Agencia Nacional de Promoción Científica y Tecnológica (PICT 2012-0421) Infoquest Foundation The Geological Society of America The Department of Earth Sciences of the University of Southern California The Austin Geological Society NSF OISE 1023978 NSF 1329471 The University of Sydney Research Collaboration Award Fieldwork and research funded by the Secretaría de Gobierno of La Rioja (to LEF), CONICET, Jurassic Foundation (to EMH), and the Agencia Nacional de Investigaciones Científicas y Técnicas (PICT 2012-0421 to LEF). Field work and research for GGT was supported by Infoquest Foundation, Jurassic Foundation, the Geological Society of America, the Department of Earth Sciences of the University of Southern California, The Austin Geological Society, and NSF OISE grant 1023978, University of Sydney Research Collaboration Award and NSF/1329471. The funders had no role in study design, data collection and analysis, decision to publish, or preparation of the manuscript.

==============================
Titanosauria is a globally distributed clade of sometimes extremely large Mesozoic herbivorous sauropod dinosaurs. On the basis of current evidence these giant dinosaurs seem to have reproduced in specific and localized nesting sites. However, no investigations have been performed to understand the possible ecological and geological biases that acted for the selection of these nesting sites worldwide. In this study, observations were performed on the best-known Cretaceous nesting sites around the world. Our observations strongly suggest their eggs were incubated with environmental sources of heat, in burial conditions. Taking into account the clutch composition and geometry, the nature and properties of the sediments, the eggshells’ structures and conductance, it would appear that titanosaurs adopted nesting behaviors comparable to the modern Australasian megapodes, using burrow-nesting in diverse media and mound-building strategies.

Introduction

Titanosaur sauropods were Mesozoic dinosaurs (Bonaparte & Coria, 1993; Wilson & Upchurch, 2003; Curry Rogers, 2005) that reached gigantic sizes (Upchurch, Barrett & Dodson, 2004; Sander et al., 2011; Benson et al., 2014) but also became dwarfed in insular ecosystems (Benton et al., 2010; Stein et al., 2010; Csiki et al., 2010). These quadrupedal herbivores are easily recognizable by their long necks and tails (Fig. 1A), small heads (Figs. 1A and 1B) and a characteristic wide-gauge stance (Fig. 1C; Wilson & Carrano, 1999; Carrano, 2005; Wilson, 2005a), a condition retained from titanosauriforms.

Figure 1 Skeletal and life restorations of titanosaur sauropods.

(A, C) Based on Futalognkosaurus dukei (Calvo et al., 2007). (B) Skull reconstructions of (up) Nemegtosaurus mongoliensis, Nowinski 1971 and (down) Tapuiasaurus macedoi Zaher et al., 2011 (based on Wilson, 2005b and Zaher et al., 2011, respectively).

Titanosaurs populated every continent including Antarctica (Curry Rogers, 2005; Mannion et al., 2011; Cerda et al., 2012) and according to Mannion & Upchurch (2010) and Mannion & Upchurch (2011) they preferred inland rather than coastal habitats. They experienced a great radiation during the Late Cretaceous, chiefly in South America, where more than 20 genera have been recorded (Upchurch & Barrett, 2005; Zaher et al., 2011; Mannion et al., 2011; García et al., 2015; Vieira et al., 2014).

Eggs and egg clutches classified within Megaloolithidae, a parataxonomic group of eggs without any modern biological principles, have been assigned to titanosaurs (as well as other taxa, e.g., Hadrosauria) (Vianey-Liaud et al., 1994; Mikhailov, 1997; Grigorescu, 2010; Grigorescu et al., 2010). Although in recent years numerous studies have explored several aspects of titanosaur reproductive biology (Cousin & Breton, 2000; Jackson et al., 2008; Sander et al., 2008; Vila et al., 2010a; Vila et al., 2010b), most of them described and compared eggs and eggshells in a non-phylogenetic context (i.e., fossil parataxonomy). Titanosaur eggs and eggshells have only been positively identified in a few instances with the discovery of embryos in ovo (Chiappe et al., 1998; Chiappe, Salgado & Coria, 2001; Wilson et al., 2010; Grellet-Tinner et al., 2011; Grellet-Tinner et al., 2012) and/or with cautious morphological characterizations of eggs and eggshells (Grellet-Tinner, Chiappe & Coria, 2004; Grellet-Tinner et al., 2006; Grellet-Tinner et al., 2011; Grellet-Tinner et al., 2012; Grellet-Tinner, Fiorelli & Salvador, 2012) that allow identification of phylogenetic characters.

Species survival is contingent on several factors: food availability, predation, competition and reproduction. Just as amniotic oviparity freed vertebrates from water-bound reproduction, viviparity entirely disconnected some environmental factors (e.g., temperature) from hatching to maximize reproductive success. Modern archosaurs lay amniotic eggs in nests and their reproduction is thus more constrained by environmental drivers than viviparous amniotes. Therefore, judicious nesting-site selection becomes a critical factor, as parents cannot compensate post-hatching for a poor choice of nesting environment (Shine & Harlow, 1996; Kolbe & Janzen, 2002; Kamel & Mrosovsky, 2005; Grellet-Tinner & Fiorelli, 2010; Grellet-Tinner, Fiorelli & Salvador, 2012). The nests of living archosaurs can be broadly divided into those that are on the ground (or below it), and those that are not, e.g., on trees and cliff faces. Although arboreal nests are quite interesting and have been the focus of several studies (Collias, 1964; Collias, 1997; Hansell, 2000; Gill, 2007; Walsh et al., 2010), including phylogenetic analyses (Winkler & Sheldon, 1993; Zyskowski & Prum, 1999; Hansell, 2007), such nests do not offer a valid model for titanosaur reproduction. Hence, this study focuses on the nests of ground dwelling species. Ground nests interface in various degrees with the sediments in which they are constructed. Among these, two types of structures are recognized based on the incubating temperature and humidity requirements (Booth & Thompson, 1991): nests in which eggs are deposited in the air-sediment interface (which are considered to be open-nests sensu Collias, 1964); and nests in which eggs are buried in the substrate (Collias, 1964; Seymour & Ackerman, 1980; Larson, 1998; Jones & Göth, 2008; Brazaitis & Watanabe, 2011).

The odds of preserving any of these nesting structures in the fossil record are extremely poor (Hasiotis et al., 2007) and/or their preservation could easily be misinterpreted. Therefore, according to the taphonomy of behavior (Plotnick, 2012), we focus this investigation not only on ichnology but also behavioral biology, offering the following revised nest definitions in the context of this study to help identify and discriminate nests from egg clutches in the fossil record:

• Archosaur nest: any recognizable structure or modification of environment that is voluntarily made by the parents to ovideposit their eggs.

• Archosaur nest function: mediates and optimizes environments in order to ensure successful egg incubation and hatching.

• Archosaur nest diagnosis: an identifiable structure recognized to result from archosaur parental nesting behavior that contains at least either autochthonous egg remains or eggshell fragments.

Herein, we review nesting sites from around the world (Fig. 2) that have been assigned to titanosaurs, their eggs, eggshell microstructures, and sedimentary data, in an attempt to understand the reasons sauropods were able to reproduce globally, but use particular localized nesting sites characterized by an overwhelming abundance of egg clutches and eggs. Understanding their nesting and incubation strategies through the megapode model might shed light on the enigmatic reproductive behavior of these extinct behemoths.

Figure 2 Upper Cretaceous paleogeography and distribution of the reviewed titanosaur nesting sites.

Map modified from Ron Blakey, Colorado Plateau Geosystems, Inc.

Materials and Methods

The fossil record of “titanosaur” eggs and shells extends to Europe, Asia, South America and Africa. However, less than 10 years have passed since their first inclusion in a phylogenetic study (Grellet-Tinner et al., 2006). In contrast, most of the earlier (e.g., Mikhailov, 1997; Mohabey, 1998; Mohabey, 2001; Garcia & Vianey-Liaud, 2001a; Vianey-Liaud & Zelenitsky, 2003) and several current studies (Fernández & Khosla, 2015; Sellés & Vila, 2015) provide descriptions lacking enough detail and/or skewed by the use of parataxonomic criteria (Grellet-Tinner et al., 2012). In addition to the egg and eggshell morphology, the study of titanosaur nesting strategies requires the combination of an array of sources of data, such as clutch geometry and spacing, description of the nesting sediments and their stratigraphic context, possible geothermal activity, availability of vegetal materials and paleoclimatic conditions.

It is unavoidable that data on some nesting sites are more comprehensive than others because some localities have been more intensively researched than others, and with different purposes (e.g., stratigraphy, parataxonomy). The need for a full and accurate record prevented the inclusion in this study of, for example, the findings of isolated eggshells in Morocco and Tanzania, in Africa (Garcia et al., 2003; Gottfried et al., 2004). In addition, it should be noted that the localities in Spain are much more restricted geographically than others in South America and Asia (Sellés & Vila, 2015). As such, the more than 220 localities with eggs and eggshells in Southern Pyrenees, (Sellés & Vila, 2015) are not directly comparable in terms of scale with other localities around the world. Although not as close together as in Spain, many of the localities reported in Southern France have also been studied for parataxonomic and biostratigraphic purposes (Garcia & Vianey-Liaud, 2001a; Garcia & Vianey-Liaud, 2001b; Vianey-liaud, Khosla & Garcia, 2003; Cojan, Renard & Emmanuel, 2003), hence not providing enough information on morphology and/or egg spatial distribution. In addition, most fossil eggs and eggshells discovered during the nineteenth and twentieth century were not recovered with appropriate stratigraphic control and/or using archaeological field techniques (Cousin & Breton, 2000). The same problem arises with the many localities reported in China and Mongolia, which yield an extensive record of megaloolithid and faveoloolithid eggs and eggshells (Carpenter & Alf, 1994; Liang et al., 2009).

Considering the above-mentioned constraints, we selected eight nesting sites distributed across four continents that provide enough data (Table S1) for comparison with modern analogues: Sanagasta (Argentina), Gyeongsang Basin (South Korea), Haţeg (Romania), Dholi Dungri (India), Rennes-le-Château and Albas (France), Coll de Nargó (Spain), and Auca Mahuevo (Argentina). It should be noted that all the selected nesting sites preserve autochthonous and/or parautochthonous fossil material in contrast to eggs and eggshells found in lithologies that belong to different environmental settings (Table 1), hence creating preservation biases.

Table 1 Egg and eggshell morphologies, spatial distribution and nesting paleoenvironments.

Sites		Sanagasta	Gieongsang Basin	Haţeg	Dholi Dungri	
Authors		Fiorelli et al., 2012; Grellet-Tinner, Fiorelli & Salvador, 2012	Huh & Zelenitsky, 2002; Kim et al., 2009.	Grellet-Tinner et al., 2012	Wilson et al., 2010	
Formation		Los Llanos	Boseong	Sânpetru	Lameta	
Age—stage		Hauterivian?- Cenomanian?	Upper Cretaceous	Maastrichtian	Maastrichtian	
Eggs	egg shape	sub-spherical	spherical	sub-spherical	spherical	
egg size (cm)	21	15–20	11–13	14–18	
eggshell thickness (mm)	1.2–7.95 (mean = 3.84)	1.33–2.2	1.7–1.8	2.26–2.36	
pore canal morphology	Y-shaped	?	Y-shaped	straight	
pore aperture morphology	round and funnel shape	?	round and funnel shape	round and funnel shape	
ornamentation morphology	nodular-single nodes and coalesecent nodes	nodular	nodular-single nodes and coalesecent nodes	nodular	
ornamentation size (mm)	Ø = 0.58–0.62	?	Ø = 0.6–0.7	?	
MT size (mm)	0.025–0.09	?	0.19	?	
Egg spatial distribution	grouping	clutches; up to 30 eggs	clutches; up to 16 eggs	clutches; average 4 eggs	isolated or in clutches; up to 12 eggs	
geometry	bowl, linear, random	?	bowl-round	bowl-round	
layers	2	1	2	?	
Paleoenvironment	sediment	medium to coarse- grained grey and whitish arkosic sands	sandy tuffaceous sandstones	fine grained siltstone- mudstone sediments	calcareous sandstones	
setting	geothermal	floodplain deposits/ geothermal/paleosols	floodplain/ paleosols/geothermal	alluvial-limnic/ paleosols	
volcanism	present	presenta	present	presentc	
paleoclimate	semiarid	semiarid	warm (MAT = 14°)	semi-aridc	
vegetation	?	C3 veg.	C3 veg.b	dominantly herbaceous; C3 veg.c	
Sites		Rennes-le-Château and Albas	Coll de Nargó	Auca Mahuevo layers 1–3	Auca Mahuevo layer 4	
Authors		Cousin & Breton, 2000; Cousin et al., 1989	Vila et al., 2010b; Vila, Jackson & Galobart, 2010	Grellet-Tinner, Chiappe & Coria, 2004; Grellet-Tinner, 2005	
Formation		Marnes rouges inférieures	Tremp	Anacleto	
Age—stage		Upper Maastrichtian	Maastrichtian	Campanian	
Eggs	egg shape	sub-spherical	spherical	spherical	spherical to sub-spherical	
egg size (cm)	16–20	20	13–15	12.5–14	
eggshell thickness (mm)	up to 2.5	1.3–4.5	1–1.78	1.7–1.8	
pore canal morphology	?	Y-shaped	straight	Y-shaped	
pore aperture morphology	round?	round-elliptical	round-elliptical and funnel shape	round and funnel shape	
ornamentation morphology	nodular	nodular	nodular-single nodes and coalesecent nodes	nodular-single nodes and coalesecent nodes	
ornamentation size (mm)	?	Ø = 0.64–0.87	Ø = 0.35–0.65	Ø = 0.68	
MT size (mm)	?	?	?	0.2–0.25	
Egg spatial distribution	grouping	isolated or in clutches; 3–8 eggs	clutches; up to 20–28 eggs	clutches; 15–50 eggs	
geometry	bowl-round/supergroups (Ø = 3, 5 m)	linear-round (inferred kidney shape)	bowl-kidney	
layers	1	2–3?	1–2	
Paleoenvironment	sediment	very soft variegated marls	mudstones intercalated with medium-coarse sandstone bodies	reddish-brown siltstones and mottled mudstonesf	
setting	floodplain?/paleosols	lagoon/marsh/fluvial/paleosols	floodplain depositsf/paleosolsf,g	
volcanism	?	?	presentg	
paleoclimate	tropical-subtropical/semi-arid episodesd	warm (MAT = 21°)/MAP = 1,200 mm/yre	warm and seasonalf dryer	warm and seasonalf wetter	
vegetation	C3 veg. riparian forest/open vegetation (Aix-en-Provence Basin)d	C3 veg.e	palustrine plant remainsf	
Notes.

a Choi, 1986; Chough et al., 2000; Choi et al., 2005; Choi et al., 2006.

b Bojar, Csiki & Grigorescu, 2010.

c Tandon et al., 1995; Tandon & Andrews, 2001; Prasad & Sahni, 2014.

d Cojan, Renard & Emmanuel, 2003.

e Riera et al., 2013.

f Garrido, 2010a; Garrido, 2010b.

g Jackson, Schmitt & Oser, 2013.

MAP mean annual precipitation

MAT mean anual temperature

MT membrana testacea

We selected Megapodiidae for modern nesting analogues. This avian family consists of 22 species that are limited to Australia, New Guinea, Micronesia, Melanesia and Polynesia (Dekker, 2007; Harris, Birks & Leaché, 2014). They are unique among living dinosaurs by exclusively using environmental heat sources rather than body heat to achieve incubation (Jones & Birks, 1992; Del Hoyo, Elliott & Sargatal, 1994).

Results

Sanagasta, Argentina

Although this site has been cautiously classified as a neosauropod nesting ground (Grellet-Tinner & Fiorelli, 2010; Fiorelli et al., 2012; Grellet-Tinner, Fiorelli & Salvador, 2012), some paleontologists have regarded eggs with similar morphology as titanosaurs (De Valais, Apesteguía & Udrizar Sauthier, 2003; Simón, 2006). Hence on this basis, we include this Cretaceous nesting site (Grellet-Tinner & Fiorelli, 2010; Fiorelli et al., 2012) in our study (Table 1). The Sanagasta nesting site located in La Rioja, Argentina (Tauber, 2007; Grellet-Tinner & Fiorelli, 2010; Fiorelli et al., 2012), represents the first account of synchronous geothermal activity and “titanosaur” nesting strategy in the fossil record (Grellet-Tinner & Fiorelli, 2010). Several clutches were recovered from a single outcrop of the Los Llanos Formation in the Sanagasta Valley (Tauber, 2007; Grellet-Tinner & Fiorelli, 2010; Fiorelli et al., 2012). The nesting sediments consist of medium to coarse-grained grey and whitish arkosic sands deposited in a hydrothermal setting (Fiorelli et al., 2012). The eggs exhibit a thick eggshell (thickness 7.9 mm–1.2 mm; mean = 3.84 mm; Grellet-Tinner, Fiorelli & Salvador, 2012) regarded as an adaptation to resist acid erosion during geothermal incubation (Grellet-Tinner & Fiorelli, 2010). The thick single structural layer of the Sanagasta eggshells displays nodular rounded structures on the outer surfaces and long eggshell units with slender concentric and convex growth lines, including secondary dichotomic branching outwards (Figs. 3A–3D). The spongy appearance of the eggshell is produced by a high concentration of wide pores that meander and branch between the eggshell units (Grellet-Tinner, Fiorelli & Salvador, 2012). Moreover, the large (∼4,800 cm3) subspherical eggs (Fig. 3E and Table 1) are perforated by ∼482,000 pores leading to a water vapor conductance of ∼2,850 mgH2O/day∗Torr (Grellet-Tinner, Fiorelli & Salvador, 2012).

Figure 3 Sanagasta nesting site.

(A, C) Radial thin sections of eggshell fragments seen under Scanning Electron Microscope (SEM) and (B, D) under stereomicroscope. (E) Complete egg (CRILAR Pv-400 SA-C6-e1). (F) Egg clutch exposed in situ at the Sanagasta Geologic Park.

Several clutches (e.g., Fig. 3F) were discovered regularly placed nearby preserved geothermal relics, encrusted with silcrete-calcrete structures, some with up to 30 eggs (Grellet-Tinner & Fiorelli, 2010; Fiorelli et al., 2012; Grellet-Tinner, Fiorelli & Salvador, 2012). Although the eggs spatial distribution in two superposed rows, where the upper row has a greater amount of eggs (Tauber, 2007; Grellet-Tinner & Fiorelli, 2010; Fiorelli et al., 2012), suggesting that they were deposited in dug out-nests, close to geothermal vents, springs, or pools, no nesting structure has been preserved (Fiorelli et al., 2013). Therefore, these fossil assemblages are referred to only as egg clutches, although the oological data (Table 1) strongly support parental placement of dug out-nests with eggs in close proximity of hydrothermal structures.

Gyeongsang Basin, South Korea

The nesting paleoenvironment of Sanagasta was compared with South-Korean nesting sites in the Gyeongsang Basin (Fiorelli et al., 2012), where continental sedimentation, volcanism and hydrothermalism occurred concomitantly during the Late Cretaceous (Choi, 1986; Chough et al., 2000; Choi et al., 2005; Choi et al., 2006). The development of this basin was associated with subduction in an Andean-type continental margin during this period (Choi, 1986). The Korean egg clutches consist of faveoloolithid eggs (Paik, Huh & Kim, 2004) similar to those found in Sanagasta. The eggs were recovered in several Upper Cretaceous localities, preserved in tuffaceous sandstones interpreted as floodplain deposits and related to meandering rivers. According to previous descriptions, the faveoloolithid eggs found in Boseong (one of the most studied localities of the Gyeongsang Basin) vary in diameter from 15 to 20 cm and have extremely porous eggshells of 1.33–2.20 mm thick (Huh & Zelenitsky, 2002; Kim et al., 2009). The taphonomic and paleoenvironmental analyses conducted by Paik, Huh & Kim (2004) and Paik, Kim & Huh (2012) revealed that several single-layered clutches of up to 16 eggs occur in different calcic paleosol levels, suggesting nesting site fidelity. Although many egg clutches were recovered, no nesting structures could be found in the encrusting sediments (Kim et al., 2009). However, Paik, Huh & Kim (2004) and Paik, Kim & Huh (2012) suggested that eggs were incubated buried in the substrate, thus laid in excavated nests. Interestingly, the geological origin and development of the Gyeongsang basin was related to subduction in an Andean-type continental margin (Choi, 1986), with epithermal deposits (Choi et al., 2005). As the latter are related to subaerial volcanism or shallow intrusions (Choi et al., 2005; Choi et al., 2006) and the nesting sediments are locally tuffaceous (Paik, Kim & Huh, 2012), Fiorelli et al. (2012) suggested that nesting in geothermal settings was not limited to a particular endemic neosauropod population but were used by allopatric populations of probably the same genus.

Haţeg, Romania

Grellet-Tinner et al. (2012) examined a group of 11 titanosaur egg clutches discovered in the exposures of the Maastrichtian Sânpetru Formation, at Toteşti, Romania (Grigorescu et al., 1990; Grigorescu, 1993; Codrea et al., 2002; Panaiotu & Panaiotu, 2010). Grellet-Tinner et al. (2012) mention the presence of geothermal-derived minerals in the eggshell pores and the presence of volcanic activity in the same basin concomitantly to the oviposition of the eggs. Moreover, a tremendous number of embryonic bones representing several taxa (titanosaurs, aves, and lepidosaurians; Grellet-Tinner et al., 2012) were recovered in the same formation, supporting possible geothermal and hydrothermal activities related to the coeval Carpathians and Apuseni volcanism (Russo-Sändulescu & Berza, 1979; Kräutner, Vajdea & Romanescu, 1986; Ştefan et al., 1988). The egg clutches were encased in fine grained siltstone-mudstone sediments, typical of a low energy nesting environment (Bojar et al., 2005; Grellet-Tinner et al., 2012). The spherical eggs were deformed by lithostatic compression, to become subspherical (Codrea et al., 2002; Grellet-Tinner et al., 2012). Their phylogenetic characters match those of the eggs from the Auca Mahuevo layer 4 (AM L#4) (Grellet-Tinner et al., 2012) that were identified as nemegtosaurid titanosaurs (García et al., 2010), based on their embryos in ovo. As such, both the AM L#4 and Toteşti eggs display similar egg shape and size, eggshell thickness, identical structural single layer of the shell, consisting of acicular calcitic crystals radiating from nucleation centers located above the membrana testacea, the same Y-shaped vertical pore canals with funnel-shaped pore apertures located between the nodular surficial ornamentation, similar unit and node shapes, as well as an horizontal pore canal network parallel to the membrana testacea (Figs. 4A–4C; Grellet-Tinner, Chiappe & Coria, 2004; Grellet-Tinner et al., 2012). The Toteşti egg clutches (Fig. 4D) consist of groups of four eggs on average that are superposed similarly to those in Sanagasta, thus suggesting they were also buried in dug-out nests (Grellet-Tinner et al., 2012). However, like the previously described nesting sites (Kim et al., 2009; Grellet-Tinner & Fiorelli, 2010), no true nesting structure was recognized. As such, the inference of nests rests only on the spatial arrangement of eggs in each of the 11 clutches (see Table 1). Grellet-Tinner et al. (2012) also mention the presence at the same site of other isolated egg species which are larger and exhibit different eggshell macro- and micro-morphologies, thus belonging at least to an additional dinosaur species. The presence of several egg types at the same site and stratigraphic horizon may indicate a preferential nesting environment for the dinosaurs confined in the Haţeg insular system. Regardless of considering the exclusive presence of sauropod (mostly titanosaurian taxa) egg clutches or mixed egg fossils including ornithopod species, although not confirmed contra Grigorescu (2010) and Grigorescu et al. (2010), the interesting question remains why only Toteşti and Nălaţ-Vad, or Tustea were preferential laying grounds on the Haţeg Cretaceous volcanic island.

Figure 4 Haţeg Basin.

(A–C) Nemegtosaurid eggshell fragments seen under SEM. (D) Egg clutches (TO O–01; IRSNB Cast-Vert 32) exhibit at the University of Cluj and the Royal Belgian Institute of Natural Sciences. From Grellet-Tinner et al., 2012.

Dholi Dungri, India

The Upper Cretaceous (Maastrichtian) Lameta Formation in India, overlain by the volcanic flows of the Deccan Traps, is well known for its rich record of dinosaur nesting sites (Mohabey, Udhoji & Verma, 1993; Mohabey, 1998). It consists of calcareous sandstones deposited by episodic transport in an alluvial-limnic paleoenvironment related to semiarid climate and secondarily affected by extensive pedogenesis (Mohabey, Udhoji & Verma, 1993; Mohabey, 1998; Mohabey, 2005; Wilson et al., 2010). Although many oospecies were recognized in the fossil parataxonomy (Mohabey, 1998), only the 3 eggs referred as Megaloolithus dhoridungriensis have been directly associated with titanosaurs, based on close association with a partial hatchling found at the exposures near the Dholi Dungri locality, Gujarat, India (Mohabey, 1998; Wilson et al., 2010).

According to Wilson et al. (2010) the spherical eggs range from 14 to 18 cm in diameter with an eggshell thickness of 2.26–2.36 mm and display discrete eggshell units similar to other megaloolithids. Although they are slightly larger and have thicker shells (Table 1), the Dholi Dungri eggs share only a few similarities with Auca Mahuevo (AM) fossils in their shape and single structural layer with acicular calcitic crystals radiating from nucleation centers located above the missing membrana testacea, a similar nodular surficial ornamentation, and a horizontal pore canal network parallel at the base of the shell with straight vertical pore canals and funnel-shaped apertures outward. However, the vertical pores do not appear to have the atypical Y-shaped ramifications and are significantly more numerous than the ones in titanosaur eggs from AM (Wilson et al., 2010). Their overall morphological characters are closer to the unidentified Toteşti and Nălaţ-Vad egg species with thicker eggshell (Grellet-Tinner et al., 2012), than the AM eggs. The Dholi Dungri titanosaur eggs have a volume of ∼2,150 cm3, an egg mass of ∼2,300 g, and appear isolated or grouped in clutches of up to 12 eggs, with an average clutch mass (6–12 eggs per clutch) of ∼21,000 g (Wilson et al., 2010).

Like other Indian sites, no nesting structures have been observed at this locality (Wilson et al., 2010), but the high porosity of the eggshells suggests burial incubation (Sander et al., 2008; Wilson et al., 2010). Tandon et al. (1995) noted that some of the different nesting sites in the Lameta beds appear to be topographically related and widely distributed in similar lithologies, suggesting a “practiced sense of site selectivity”.

Rennes-le-Château and Albas, France

Since the late 1800s, several titanosaur egg clutches have been discovered in Cretaceous exposures in Southern France (Freytet, 1965; Kerourio, 1981; Cousin et al., 1989; Cousin et al., 1994). However, according to Cousin & Breton (2000), the vast majority of the fossils were recovered with inadequate field techniques and/or inappropriate stratigraphic control. Therefore, we focus on the detailed excavations of the nesting sites at the Upper Maastrichtian, Rennes-le-Château (Cousin et al., 1994) and Albas (Cousin & Breton, 2000), as these two sites were quarried with archaeological techniques, thus providing a good overview of the taphonomy and nesting environment (Cousin & Breton, 2000).

The eggs typically exhibit eggshell thicknesses of up to 2.5 mm, although some samples show thinning by dissolution (Cousin & Breton, 2000). As in other titanosaur eggs, the French eggs display nodular rounded structures on their outer surfaces and spherulitic eggshell units with slender concentric and convex growth lines formed by acicular calcitic crystals radiating from nucleation centers located above the missing membrana testacea. Moreover, they display a horizontal pore canal network, parallel to the base of the shell, with straight vertical pore canals for gas conductance (some of them with complex branching network of secondary transverse canals and dichotomic Y-shaped ramifications), and pore apertures around the base of the external nodes. These eggshells are quite similar to those of titanosaur “megaloolithid” eggshells from Spain which, according to Jackson et al. (2008), have a high pore density and elevated gas conductance, nearly 4,000 mgH2O/day∗Torr (an overestimate due to a calculation error for gas conductance of the eggshells by these authors). The 17–20 cm eggs have a volume of ∼2,100 cm3, an egg mass of ∼2,300 g and a clutch mass varying from ∼18,500 g (Rennes-le-Château) to ∼35,000 g (Albas). In Rennes-le-Château the eggs were found isolated or in small arcuate rows, but also in clusters of 3–8 eggs (Cousin et al., 1989). The careful excavation revealed that the egg clutches were preserved in situ (Cousin & Breton, 2000; Cojan, Renard & Emmanuel, 2003) and deposited in different stratigraphic levels, thus, suggesting nesting site fidelity (Cousin et al., 1989; Cousin & Breton, 2000). The sedimentary evidence does not support the presence of true nests (as above-defined), although Cousin & Breton (2000) suggested that the eggs of the Albas clutch could have been deposited in shallow pits. Therefore, the lack of any supporting sedimentary structure does not allow the inference of sauropod nests at these two sites. Although the egg clutches from Southern France do not indicate any organic matter in the surrounding sediments, the possibility of incubation in nests with vegetal mounds has been suggested by Kerourio (1981) and Cousin & Breton (2000), based primarily on high conductance of the eggshells. Yet, Grellet-Tinner, Fiorelli & Salvador (2012) suggested that high conductance values alone do not imply incubation in mounds, but just environments with elevated moisture contents.

Coll de Nargó, Spain

Several sauropod egg clutches were reported in Northeastern Spain (Vila et al., 2010a; Vila et al., 2010b; Vila et al., 2011; Vila, Jackson & Galobart, 2010), a region that was also tectonically active at the time the eggs were oviposited (Puigdefàbregas, Muñoz & Vergés, 1992). These fossils were classified in the Megaloolitidae oofamily and later loosely associated with titanosaurs (Sander et al., 2008; Vila et al., 2010a; Vila et al., 2010b) after the discovery of titanosaur embryos in the eggs of Megaloolithus patagonicus (Chiappe et al., 2003), and M. dhoridungriensis (Wilson et al., 2010). According to Sellés et al. (2013), more than 30 egg-bearing stratigraphic levels distributed in two lithofacies, representing a fluvial paleoenvironment, are recognized in the Upper Cretaceous Tremp Formation (Sander et al., 2008; Vila, Jackson & Galobart, 2010; Vila et al., 2010b). Both lithofacies and eggs show evidences of stress deformation, consistent with the tectonic stress orientation of the region (Vila et al., 2010b).

The eggs and eggshells have been described by several authors (Jackson et al., 2008; Vila et al., 2010a; Vila, Jackson & Galobart, 2010; Sellés et al., 2013). They are spherical and reach 20 cm in diameter (Vila, Jackson & Galobart, 2010). Their 2.5 mm thick eggshells show typical discrete units with slender concentric and convex growth lines of acicular calcite crystals radiating from the nucleation centers, as well as a pore network system with branching—dichotomic Y-shaped ramifications—and secondary transversal ones.

According to Sellés et al. (2013) the 75 clutches found in situ at Pinyes (a subsite at Coll de Nargó locality) support a nest site fidelity behavior. However, Vila, Jackson & Galobart (2010) indicated that these clutches represent a single event, albeit no sedimentary evidence independently confirms their interpretation (Sander et al., 2008). Three types of egg clutches were recognized by Vila, Jackson & Galobart (2010). The “type 1”, consists of clutches with 20–28 eggs separated by 3–6 m and buried in bowl kidney-shaped depressions (Vila, Jackson & Galobart, 2010; Vila et al., 2010b). However, in previous interpretations these were considered as superimposed clutches with fewer eggs (Peitz, 1998; Sander et al., 1998; Sander et al., 2008). According to Vila et al. (2010a), Vila et al. (2010b) and Vila, Jackson & Galobart (2010) but contra Sander et al. (2008), the egg spatial arrangement in these putative complete clutches coupled with the high water vapor conductance (GH2O) of the eggshells (Deeming, 2006), would suggest the eggs were buried during incubation.

Auca Mahuevo, Argentina

This Campanian locality (Dingus et al., 2000) in the Anacleto Formation (Argentina) was the site from which the first reported titanosaur embryonic bones and soft tissues in ovo were discovered (Chiappe et al., 1998; Chiappe, Salgado & Coria, 2001; Salgado, Coria & Chiappe, 2005). They were recently re-identified as nemegtosaurids (García et al., 2010). The eggs are in four egg-bearing strata (Chiappe et al., 2003; Chiappe et al., 2004) consisting of reddish-brown siltstones and mottled mudstones (Chiappe et al., 2000; Chiappe et al., 2004; Garrido, 2010a), deposited in an alluvial plain (Sander et al., 2008; Garrido, 2010a) under the regime of a warm and seasonal climate (Garrido, 2010a). The eggs and eggshells were described in detail by Grellet-Tinner, Chiappe & Coria (2004) and Grellet-Tinner (2005). According to these authors the well-preserved eggshell specimens, averaging 1.30 mm thick, display a pronounced ornamentation of single nodes and a pore network that consists of vertical channel openings in funnel-like structures located between the surficial nodes (Figs. 5A and 5B). Some vertical pores branch in a “Y” pattern, a derived feature originally described for these eggs but shared with other titanosaurs (Figs. 5C and 5D; Grellet-Tinner, Chiappe & Coria, 2004; Grellet-Tinner, Fiorelli & Salvador, 2012). Additionally, they show horizontal canals located between the bases of the eggshell units—nucleation centers—and above the membrana testacea (Figs. 5C–5D). The unhatched eggs range from 13 to 15 cm in diameter (Fig. 5E; (Chiappe et al., 1998; Grellet-Tinner, Chiappe & Coria, 2004)) and are preserved in clutches of 15 to 40 eggs (Fig. 5F; Chiappe et al., 2000; Grellet-Tinner, Chiappe & Coria, 2004). Although supposedly in their original position, egg clutches became undistinguishable from one another due to soft sediment deformations (G. Grellet-Tinner (or GGT), 1999–2000; Chiappe et al., 2003; Jackson, Schmitt & Oser, 2013). Like the above-mentioned European, Asian, and South American nesting sites, the dense accumulation of clutches (11 eggs/m2; Sander et al., 2008) in egg layers 3 and 4, led to interpretations of gregarious and nesting site fidelity behaviors (Chiappe et al., 2000; Chiappe et al., 2003). Paradoxically despite the extent of this nesting site, no nesting structures were reported in these overbanking sedimentary layers which were, according to Garrido (2010a), the preferred laying grounds close to the stream channel for these nemegtosaurids. However, six trace fossils interpreted as nests with accumulations of eggs were reported in alluvial deposits (Chiappe et al., 2004) of an abandoned channel in the AM L#4 (Chiappe et al., 2003; Sander et al., 2008), which clearly contradict “the preferred areas close to stream channels” of Garrido (2010a). The six rimmed, sub-circular to kidney-shaped structures contain 4–35 randomly disposed eggs in 1 or 2 superposed rows. They were interpreted as rimmed-nests, ranging from 85 to 125 cm, and 10 to 18 cm deep (Chiappe et al., 2004; Sander et al., 2008). The six fossil assemblages were regarded as nests and an “open nest” strategy was suggested for the entire, assumed monospecific nesting site (Chiappe et al., 2004; Sander et al., 2008). However, a recent re-evaluation of the putative nests concluded that the rimmed structures were titanosaur footprints in an abandoned channel (Grellet-Tinner, Fiorelli & Salvador, 2012) that randomly trapped eggs during the several episodic floods. This interpretation is consistent with all the geological data and supported by the inconsistency of an “open nest” hypothesis (Chiappe et al., 2004; Jackson et al., 2008; Sander et al., 2008), considering the high GH2O of the eggshells (Grellet-Tinner, Fiorelli & Salvador, 2012). Like other titanosaurs, the AM eggs’ morphology indicates they were likely incubated in relatively high nesting humidities (Grellet-Tinner, Chiappe & Coria, 2004; Grellet-Tinner et al., 2006).

Figure 5 Auca Mahuevo nesting site.

(A–C) Radial section of titanosaur eggshell fragments (From Grellet-Tinner, Chiappe & Coria, 2004; Grellet-Tinner & Zaher, 2007). Note in (C) the transverse Y-shaped vertical pore canals (blue arrow) and the horizontal network system (red arrow). (D) Schematic interpretation of an eggshell from AM L#4, according to Grellet-Tinner, Chiappe & Coria (2004) SEM observations (modified from Grellet-Tinner, Fiorelli & Salvador, 2012). a, arteries; b, branches of pore canal; c, capillaries; cm, corioallantoid membrane; eu, eggshell unit; hpc, horizontal pore canals; mt, membrana testacea; n, node on outer eggshell surface; pa, pore aperture; pf, protein fibers of the membrana testacea; v, veins; vpc, vertical pore canals. (E) Complete egg AM L#3. (F) Egg clutch recovered from AM L#3 (LACM 149648; from Grellet-Tinner, Chiappe & Coria, 2004). LACM, Natural History Museum of Los Angeles County.

Discussion

Titanosaur nesting strategies

The taphonomy of behavior emphasizes the need for defining trace fossils in terms of both classical ichnology and modern behavioral biology (Plotnick, 2012). Furthermore, the concept of behavioral fidelity, expressed as the “extent to which trace fossils preserve original behavioral signals” (Plotnick, 2012), allows behavioral interpretations contrastable to currently existing data. Considering solely the sedimentological evidence, none of the above-mentioned nesting sites display a diagnostic fossil nest structure. In the absence of direct evidence of preserved fossil nests, eggs and their spatial grouping, eggshell morphologies, coupled with observations of surrounding sediments, provide the only data with respect to nesting moisture content and heat, as those are the main extrinsic parameters that are paramount for hatching success. Pore canals allow the diffusion of gases and water vapor through the eggshell (Paganelli, 1980). Their size, geometry, and number reflect a specialization to the habitat where nesting occurs (Williams, Seymour & Kerourio, 1984; Cousin, 1997; Grellet-Tinner, Fiorelli & Salvador, 2012). Eggshell ornamentation (or lack thereof) is another morphological characteristic from which a nesting paleoenvironment can be hypothesized. Cousin (1997) and Cousin & Breton (2000) used the characteristics of the nodular appearance of eggshells from the Late Cretaceous of France as an indicator of the substrate, or nesting material, surrounding the eggs. This character (Grellet-Tinner, Chiappe & Coria, 2004; Grellet-Tinner & Zaher, 2007; Grellet-Tinner et al., 2011) was regarded as a specialization, increasing gas conductance through the pores that are located around and at the base of each node, by preventing nesting debris from plugging their apertures (Sabath, 1991). The densely packed nodular structures typically observed on the outer surfaces of pristine titanosaur eggshells provide a substantial increase of surface in contact with the surrounding environment, suggesting they could also have acted to buffer acidic erosion during long periods of incubation, an interpretation consistent with GGT and colleagues’ (G Grellet-Tinner, 2014, unpublished data) observations on modern megapode eggshells and associated nesting environments in Australia. Gas diffusion occurs through the pores and can be quantified by the eggshell’s GH2O (Seymour, 1979). It is commonly obtained for modern birds and reptiles by measuring water loss in a known vapor gradient across the shell (Ar et al., 1974). GH2O for fossil eggs can be estimated from eggshell thickness and pore-system geometry and so forms a valuable proxy for assessing the moisture content in dinosaur nests, their environments (Seymour, 1979; Seymour & Ackerman, 1980), and nesting strategies (Ar et al., 1974; Seymour, 1979; Birchard & Kilgore, 1980; Seymour et al., 1987; Grellet-Tinner, Chiappe & Coria, 2004; Deeming, 2006; Grellet-Tinner, Fiorelli & Salvador, 2012).

Titanosaurs could not have used the classic contact incubation strategy typical of most modern dinosaurs (Seymour, 1979; Werner & Griebeler, 2011; Ruxton, Birchard & Deeming, 2014), thus must have relied on external environmental heat for incubating their eggs. Interestingly, the extant Australasian megapodes are one of the most intriguing avian dinosaur families because they exhibit a practice unique among modern birds, regarded as a reversal character, of incubating their eggs by utilizing only environmental heat sources rather than body heat (Booth & Thompson, 1991; Jones & Birks, 1992; Del Hoyo, Elliott & Sargatal, 1994). Moreover, they display the most diversified incubation behaviors among ground nesting archosaurs (Jones & Birks, 1992; Del Hoyo, Elliott & Sargatal, 1994; Harris, Birks & Leaché, 2014), which are associated with species specific nesting strategies and nesting sites (Boles & Ivison, 1999; Harris, Birks & Leaché, 2014). These are: (1) mound-building; (2) burrow-nesting using geothermal heat; (3) burrow-nesting using solar-heated beaches; (4) burrow-nesting using decaying tree roots; (5) mound parasitism. As such, Megapodiidae potentially offers a valid proxy for titanosaur reproductive behaviors.

Among the titanosaur nesting sites reviewed in this investigation, only two have been positively related with geothermalism: the well-documented Sanagasta and the South Korean eggs (Table 1). The compact arrangement of the Sanagasta eggs, in clutches of one or two layers, led Grellet-Tinner & Fiorelli (2010) and Fiorelli et al. (2012) to suggest incubation in excavated nests, a strategy already inferred by Paik, Huh & Kim (2004) and Paik, Kim & Huh (2012) for the eggs from Seonso Formation (Gyeongsang Basin, South Korea) (Table 1). The eggs from these sites have a similar morphology (but a thinner eggshell thickness in Gyeongsang) and they also share sedimentary and geological features of geothermal activities concomitant with the oviposition. This supports the hypothesis that geothermal heat sources could have been used by certain species of Cretaceous titanosaurs. Several megapode species—Macrocephalon maleo Muller, 1846, Eulipoa wallacei Gray, 1861 and many Megapodius spp.—use burrow nesting with vegetal decomposition, solar radiation and/or geothermalism (inclusive here of volcanism) as incubating strategies (Frith, 1956; Dekker & Brom, 1960; Jones & Birks, 1992; Del Hoyo, Elliott & Sargatal, 1994; Göth & Vogel, 1997; Dekker, 2007; Bowen, 2010; Harris, Birks & Leaché, 2014). The remarkable instance of opportunistic nesting in geothermal settings represents an adaptive case in which species avoid thermally heterogeneous nesting environments by exploiting geothermal conditions that maintain higher and more constant temperatures and moisture levels in egg clutches (Werner, 1983; Göth & Vogel, 1997; Chen, Kam & Lin, 2001; Wu & Kam, 2005; Guo et al., 2008; Huang et al., 2009; Sas, Antal & Covaciu-Marcov, 2010; Grellet-Tinner & Fiorelli, 2010; Grellet-Tinner, Fiorelli & Salvador, 2012). For example, the Malau megapode (Megapodius pritchardii Gray, 1864) digs pits more than 2 m deep to utilize underground geothermal heat (Frith, 1956; Del Hoyo, Elliott & Sargatal, 1994; Göth & Vogel, 1997). In addition, because burrow nests in geothermal ecosystems are less susceptible than mounds to predation, burrow-nesting megapodes can abandon their nests after burying their clutch (Dekker, 1989; Del Hoyo, Elliott & Sargatal, 1994), a strategy consistent with the titanosaur behavior inferred from the fossil record (Sander et al., 2008; Sander et al., 2011; Werner & Griebeler, 2011; Ruxton, Birchard & Deeming, 2014), where nesting sites were often located and synchronous with geothermal activities. Paradoxically, megapode nest-burrows are dug preferably in soft soils, in volcanic sands, environments that inhibit preservation of such structures in the fossil record because they easily collapse (Frith, 1956; Dekker & Brom, 1960; Roper, 1983; Bowen, 2010).

Although, the first reports of AM nesting site implied a monotaxic titanosaur assemblage, further refined to nemegtosaurid titanosaur (García et al., 2010), one of the co-authors (Eagle et al., 2015) has determined that the egg-laying titanosaurs in AM L#4 may represent a different nemegtosaurid species, certainly closely related to those nesting in Auca Mahuevo layers 1–3 (AM L#1–3) but displaying sufficient autapomorphies to justify a species variation (Table 1). This species variation is also supported by an environmental change. Celestite geodes and barite are ubiquitous in AM L#1–3 (Garrido, 2010a; Garrido, 2010b). These two minerals are readily produced in geothermal and evaporitic settings, both equally possible at AM due to its particular geology (Jackson, Schmitt & Oser, 2013). Moreover geochemical analyses reveal a higher concentration of magnesium and lithium in AM L#1–3 than AM L#4 (Eagle et al., 2015). These two minerals are prevalent in continental brines, which could also be formed under high evaporitic conditions or geothermalism. However, the evidence available suggests AM have been selected first by a certain nemegtosaurid species (AM L#1–3) for its presence of limited rivers in a semiarid environment and then, after a climatic change toward wetter conditions, replaced by another closely related species with more conspicuous nodular eggshell ornamentation, adapted to a more humid nesting environment (Table 1). Climatic and environmental changes are also documented in the clay fabric in the 4 AM layers (Jackson, Schmitt & Oser, 2013). Interestingly, among the modern Megapodiidae a similar niche partition is observed in Australasia. The mound-builder Alectura lathami Gray, 1831 (brush-turkey) nests in wetter environments in coastal Australian regions with respect to its congener Leipoa ocellata Gould, 1840 (malleefowl) that nests in semi-arid settings. Although both species are contemporaneous their geographical niche partitioning is related to vegetation and climatic differences. Such species specific nesting partitioning in modern Australia may explain the nemegtosaurid successive species replacement in AM from a dryer environment nesting adaptation, such as occurs in AM L#1–3, to a wetter setting, recorded upwards by the transition to the Allen Formation’s estuarine-coastal sediments (Garrido, 2010a; Garrido, 2010b). Combined with the lack of convincing geothermal evidence, although not entirely discounted due to the episodic explosive volcanism (Jackson, Schmitt & Oser, 2013), the horizons with egg clutches and the eggshell structures would suggest mound-building nesting strategies with a dryer climate for AM L#1–3 which display shallower nodular eggshell ornamentation and wetter for AM L#4 with pronounced nodular ornamentation (Table 1). The oological material from AM L#4 and Haţeg are very similar, sharing several synapomorphies including egg size, shape and eggshell microstructure (Grellet-Tinner, Chiappe & Coria, 2004; Grellet-Tinner et al., 2012). The complexity of their pore systems is consistent with a morphological adaptation to high moisture nesting environments, typical of burial conditions. Furthermore, in AM a semi-arid dry/wet climate coupled with episodic volcanism contributed to vertisol horizons development at the floodplain areas (Garrido, 2010a; Jackson, Schmitt & Oser, 2013). Rhizoliths and root traces as well as small fossil logs found there (Garrido, 2010a; Garrido, 2010b; Jackson, Schmitt & Oser, 2013) suggest a floristic abundance that would promote ideal conditions for mound-nesting behaviors.

Although crocodilian mound-nests also average 1 m high and 3m in diameter (Joanen, 1969; Webb, Messel & Magnusson, 1977; Seymour & Ackerman, 1980; Waitkuwait, 1989), it is important to distinguish this type of vegetal mounds from those of the Australian brush-turkey megapodes, built from humus, soil and smaller amounts of true vegetal matter. Modern megapodes construct surprisingly large nest-mounds, relative to their egg and clutch sizes. Although a few mounds could reach 4 m in height, 18 m in length and 5 m in width, a typical brush-turkey mound commonly measures 1 m by 4 m (height and diameter respectively) and requires up to 5 tons of soil mixed with vegetal matter (Seymour & Ackerman, 1980; Jones & Birks, 1992; Del Hoyo, Elliott & Sargatal, 1994; Harris, Birks & Leaché, 2014). Conversely, nesting structures smaller than 0.75 m high and 2 m long are not functional (Jones & Birks, 1992; Del Hoyo, Elliott & Sargatal, 1994; Jones & Göth, 2008). In contrast, malleefowl mounds consist mostly of sandy material with terrigenous clasts. Malleefowl build their mound, and then dig out the center into which semi-arid to arid vegetation, like spinifex, is introduced before the rainy season (Frith, 1959; Jones & Birks, 1992; Jones & Göth, 2008; D. Booth, pers. comm., 2013). Eggs are laid around this center core that provides sufficient heat from vegetal decomposition to support embryonic development. This results in all eggs being placed relatively centrally within a mound (D. Booth, pers. comm.). Temperatures in the mound range from 27 to 38 °C, although eggs mostly lie in the range of 32 to 36 °C (Booth, 1987). In contrast, brush-turkey females burrow into the mound obliquely. The litter material does not collapse around the tunnel as sand would in a malleefowl mound (D. Booth, pers. comm., 2013). Hence, eggs can be dispersed throughout the place in large mounds and not just in the central core as in the malleefowl ones. One strategy would result in eggs grouped together in a compact clutch, while the other results in eggs isolated or lined up in small groups when the mound is eroded away. In comparison, the latter matches the pattern already observed in the titanosaur nesting sites of Southern France, where Cousin et al. (1989) and Cousin & Breton (2000) interpreted that small groups of eggs (2–4) are not randomly distributed but belong to circular “supergroups” of up to 15 eggs and 3.5 m in diameter (see Fig. 10 in Cousin & Breton, 2000), substantially separated from others (Cousin et al., 1989; Cousin & Breton, 2000). Additionally, the few-egg clutches found in close association in Coll de Nargó that were interpreted by Vila, Jackson & Galobart (2010) as partially preserved ∼ 25-egg clutches, show similar patterns. As such, the egg spatial distribution, in small clusters linearly to compactly grouped, but contained in round shaped areas of up to 2.3 m (see Table 1; Vila et al., 2010b), would either support burrow- or mound-nesting (Cousin & Breton, 2000).

The Upper Cretaceous eggs and eggshells found at the reviewed nesting sites of Southern France and Northern Spain share many characteristics. Regarding their similarities (Table 1), and considering the high conductance values of the Pinyes’ eggs, a highly humid incubation environment can be inferred for both locations. Although the evidence at hand suggests burial incubation, no record of organic matter or hydrothermal relics has been identified in their fine surrounding sediments. Regarding the morphological aspects, the minimal differences in size between eggshell pores and the fine sediments reported in both cases obscure the inference of vegetation as nesting material.

Although there are no published analyses on gas conductance for the Dholi Dungri eggs, the high values obtained from other Indian localities, ranging between 2,650 mgH2O/(day∗torr) and 3,490 mgH2O/(day∗torr) (Sahni et al., 1994), suggest a high moisture nesting microenvironment. Considering the nodular appearance of the eggshells as a good indicator of the surrounding nesting material (Cousin & Breton, 2000) the provision of substantial amounts of plant debris as constructing material cannot be discarded. Although the sediment itself is coarse enough to prevent pore obstruction, the well-developed ornamentation of the eggshells could be a mechanism to prevent the external eggshell erosion, by the acid formed by decomposing microbes and fungi during an extended incubation period, like in modern megapodes. The morphological and sedimentological evidences at hand, coupled with the interpretation of a semi-arid, tropical dry-wet climate for the Lameta Formation during Maastrichtian (Tandon et al., 1995; Tandon & Andrews, 2001; Wilson et al., 2010; Prasad & Sahni, 2014), allows us to infer an A. lathami-like nesting strategy, similar to that suggested for AM L#4 (Table 1). However, regarding the magnitude of the volcanism responsible of the Deccan Traps deposition, during Late Cretaceous, the geothermalism could also be regarded as an equally plausible alternative heat source for egg incubation. Interestingly, although Tandon et al. (1995) suggest the nesting activity in the Lameta Formation was contemporaneous with the first Deccan lavas, questions regarding its possible relationship still require further research.

Many species of modern megapodes are known to use mound-nesting strategies to incubate their eggs (Harris, Birks & Leaché, 2014). However, they are also known to revert to less conspicuous nesting strategies such as burrow-nesting (Dekker, 1989) because of their behavioral lability (Harris, Birks & Leaché, 2014). As such, although dusky megapode (Megapodius freycinet Gaimard, 1823) populations are categorized as mound-builders, they are also burrow-nesters and exploit geothermal resources for heat and moisture in New Britain and the Solomon Islands (Roper, 1983). This species demonstrates an interesting behavioral plasticity (Frith, 1956), yet still relying on environmental heat only. Additionally, the Moluccan megapodes (Eulipoa wallacei) lay their eggs in black volcanic sands and silt utilizing solar heat (Del Hoyo, Elliott & Sargatal, 1994; Dekker, 2007) but are also known to switch to geothermal and microbial decomposition. Because of the scarcity of suitable nesting areas, burrow-nester megapodes (e.g., M. freycinet, M. maleo, and M. pritchardii) commonly adopt colonial nesting behaviors and site fidelity (Del Hoyo, Elliott & Sargatal, 1994). Similarly, strong nesting environment selectivity and colonial nesting were related to the massive occurrence of eggs and egg clutches in similar lithofacies, in some of the best known titanosaur nesting sites (Sahni et al., 1994; Mohabey, 2001). Although some titanosaur species may have adopted nesting strategies relying on various styles of mound-nests, it is entirely plausible that a few of them might have reverted to such opportunistic behaviors when available, particularly in settings like Haţeg (Grellet-Tinner et al., 2012), India (Tandon et al., 1995) and AM (Jackson, Schmitt & Oser, 2013) where volcanism and related processes were coeval with nesting titanosaurs (Table 1).

Seymour & Ackerman (1980) emphasized that vegetal decomposition through microorganisms could be critical in terms of gas tension for mound-builder archosaurs. When clutches are large, as occurs with the ∼20 egg clutches of the Australian brush-turkey, the gas tensions as well as the temperature in the mound vary with the reworking of the mound by parents (Seymour & Ackerman, 1980; Del Hoyo, Elliott & Sargatal, 1994; Priddel & Wheeler, 2003). Assuming similar incubation strategies, it seems that the sizes of the sauropods could have prohibited active reworking of the mound to compensate for gas and temperature variations. Yet, such post-oviposition activities might have not been required, e.g., in modern alligators, that exhibit similar “en masse” laying behaviors.

Conclusion

Titanosaurs reproduced in globally distributed but highly particular and localized nesting sites (Sahni et al., 1994; Mohabey, 2001). The geological-sedimentological record of the reviewed titanosaur sites suggests that different titanosaur species may have evolved an array of reproductive strategies comparable to those of modern megapodes. The observed eggshell features indicate high nesting moisture content, yet with variable nesting humidities as expressed by the geometry of the pore canals used for gas diffusion through the eggshells. The egg spatial positions in clutches suggest excavated holes (e.g., Fig. 6A) as well as mound incubation (Fig. 6B). Mound-nesting incubation in its various forms as observed in modern megapodes, although not entirely supported because of the absence or paucity of organic matter or plant debris in close association with all the eggs or egg clutches in the fossil record, may be quite plausible. The sites—Haţeg, Dholi Dungri, Rennes-le-Château, Albas, Coll de Nargó and AM—seem to support mound incubating, but tectonism associated with volcanism was recorded in these sites, thus leaving an open door for opportunistic geothermal incubation strategies. In that regard geothermal and hydrothermal settings (Fig. 6) have undoubtedly been exploited by in Sanagasta and the South Korean southern peninsula for their external heat and moisture.

Figure 6 Schematic reconstruction of different nesting environments and the nesting strategies suggested for the Cretaceous titanosaur dinosaurs.

(A) Detail of borrow-nesting in geothermal environments. (B) Detail of mound-nesting and eggs buried in a soil profile.

Supplemental Information

Table S1 List the of known Cretaceous “titanosaur” egg-bearing localities/areas that preserve at least complete eggs

Nesting sites selected for the current study correspond to the localities/areas that are highlighted in grey. AL, allochthonous; AU, autochthonous; PA, parautochthonous; X, known data; ?, imprecise information; -, unknown data.

Click here for additional data file.

We acknowledge the help of Secretaría de Cultura de La Rioja. We would also like to thank the CRILAR staff and technicians (Sergio de la Vega and Carlos Bustamante) for their help and support during our work. EMH would like to acknowledge Dr. Ron Blakey for providing the Cretaceous paleogeographic map.

Additional Information and Declarations

Competing Interests

Author Contributions

The authors declare there are no competing interests.

E. Martín Hechenleitner analyzed the data, contributed reagents/materials/analysis tools, wrote the paper, prepared figures and/or tables, reviewed drafts of the paper.

Gerald Grellet-Tinner analyzed the data, contributed reagents/materials/analysis tools, wrote the paper, reviewed drafts of the paper.

Lucas E. Fiorelli analyzed the data, contributed reagents/materials/analysis tools, prepared figures and/or tables, reviewed drafts of the paper.

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
