# Peer review of "What do giant titanosaur dinosaurs and modern Australasian megapodes have in common?"

_PeerJ, doi:10.7717/peerj.1341_

## Round 0.1 · original submission · Major Revisions

Dear authors,

As the two reviewers have made differing recommendations, I have accepted the decision of 'major revision'. Please pay close attention to the recommendations of reviewer two.

Reviewer 1 ·

Basic reporting

The paper is a summary of the nesting sites characteristics attributed to Titanosaurs drawing inferences no the physical conditions likely present. This information is then compared with that of the extant megapode birds as a modern day example of what these large dinosaurs were doing with their nests.

The paleo descriptions seem adequate (my expertise lies outside this area). The one comment that comes to my mind that it is never made clear what the advantages of taking advantage of geothermal or heat of decomposition might be. For example phenotype of reptile hatchlings is know to be influenced by the environmental temperatures experienced during incubation. This could be a reason for a more specific nest selection for thermal environment. Another possible reason might be related to the probably lengthy incubation period of such large eggs. See Ruxton et al 2014 Biology letters suggested that nesting strategies may have been modified in large sauropods to reduce the chances of egg predation during incubation.

Experimental design

This is primarily a summary of nest characteristics and then a correlation is drawn with an extant group of species. In the context of paleontological research the design is appropriate.

Validity of the findings

I find the comparison of dinosaur nesting with megapodes interesting and consistent with previous work (e.g. Seymour). The hypotheses drawn are reasonable from a paleontological perspective of inference based on modern animal models.

Additional comments

The text in the manuscript needs some reworking. It is clear the authors native language is not English. I have done some marking but as with all such edits may require additional modifications. I suggest finding a reader to check verbage before submitting revised text.

Annotated reviews are not available for download in order to protect the identity of reviewers who chose to remain anonymous.

·

Basic reporting

General comments

This article presents an interesting review of some of the best preserved titanosaur eggs/nest sites. It argues that titanosaurs probably utilised dug-out nests and mound-building in a manner similar to extant megapode birds. This seems to be a plausible idea that is supported by the evidence that is presented. I believe, however, that there are some problems with the current paper that need to be addressed before it is ready for publication.
First, and most minor, there are quite a few typographical errors, missing words, and other presentational problems. i have listed many of these below, but I cannot be sure that i have found all of them. So, the paper will need careful checking and proof-reading.
Second,, and more serious, this paper is about titanosaur sauropods, but it actually says very little about these animals (aside from the details concerning nesting, eggs etc.). If I were not already familiar with these dinosaurs, all I would have learned about them from this paper is that they were globally distributed Cretaceous sauropods. Surely the paper needs to provide more background on titanosaurs, such as the fact that they were long-necked herbivores, include the largest known terrestrial animals (though there were also several small or dwarf forms), had a wide-gauge stance, apparently displayed a preference for inland habitats and so on. I therefore think the authors should add one or two paragraphs on titanosaurs to their Introduction (and perhaps a figure showing a skull and skeletal reconstruction), so that the reader can place the details concerning eggs/nests in the wider context. This should cite at least some of the recent general papers on titanosaurs - I list some appropriate references at the end of this review.
Finally, there is also an important methodological issue that needs attention. The paper does not have a ‘Materials and Methods’ section, and I get the impression that the authors have simply selected several of the ‘best’ titanosaur nest sites for review. There is no quantitative/statistical analysis of the data (perhaps because sample sizes are relatively small), and no consideration of possible sampling biases. The danger is that there is a sampling bias in the fossil record that might skew our knowldge of titanosaur nesting behaviour. Titanosaurs might have laid their eggs in all sorts of different ways and in all sorts of different environments, but perhaps only a subset of these behaviours/environments promoted particularly good preservation. i would like to see a more general overview of the titanosaur egg/nest record (I am not asking for a detailed review of everything, but a tabulation of all sites or instances where titanosaur egg remains have been found would do). the importance of this issue can be illustrated by a hypothetical example. If, for the sake of argument, the authors review four sites, the results might well be meaningful if there are only six sites known globally. If, on the other hand, they have selected four sites out of 100, then the implications of their work are less secure. So, there needs to be some discussion of the titanosaur fossil record generally, their egg and nest record in particular, and some explanation of the rationale used to select the egg/nest sites that are examined in detail. A quantitative approach would be nice (e.g. a statistical analysis of whether titanosaur eggs/nests occur more or less often in certain habitats than do other sauropod, or other dinosaur, eggs), but even a qualitative overview of these issues would allow the reader to gauge to what extent the selected nest sites are, or are not, representative of the titanosaur fossil record as a whole.
Once the issues outlined above have been resolved, I would welcome publication of this work.

Specific points

Lines 20-22 - ‘…eggshells structures and conductance, it would appear that titanosaurs have adopted the same labile nesting behaviors than the modern Australasian megapodes, using burrow-nesting in diverse…’

‘same labile nesting behaviors than the modern’ should be ‘same labile nesting behaviors as the modern’.

Lines 47-48 — ‘ …on ichnology but also behavioral biology, offering the following revised nest definitions in the context of this study to help identifying and discriminating nests from egg clutches in the fossil record’

‘to help identifying and discriminating’ should be ‘to help identify and discriminate'

Lines 50-51 - ’Archosaur nest: Any recognizable structure or modification of environment that is voluntary made by the parents to ovideposit their eggs.’

‘voluntary’ should be ‘voluntarily'

Lines 69-71 - ‘…Grellet-Tinner, Fiorelli & Salvador, 2012), other paleontologists have previously regarded eggs with similar morphology than the Sanagasta specimens as titanosaurs (De Valais, Apesteguía & Udrizar Sauthier, 2003; Simón, 2006). Hence on this basis, we include this Cretaceous nesting site…’

‘with similar morphology than the Sanagasta’ should be ‘with similar morphology to the Sanagasta'

Results section generally - In the results section, the authors discuss a series of titanosaur nesting sites/egg remains. Although geological formations are noted, the authors sometimes do not give the ages of these formations (e.g. Hatteg). It would be helpful is such ages could be mentioned (perhaps formation and age data could also be added to Table 1?).

Lines 154-155 - ’Sanagasta, thus suggesting they were also buried in dug-out nests (Grellet-Tinner et al., 2012). Yet, alike the previous nesting sites (Kim et al., 2009; Grellet-Tinner & Fiorelli, 156 2010), no true nesting…

‘Yet, alike the’ should be ‘Yet, like the'

Lines 200-201 - ‘Since late 1800s, several titanosaur egg clutches were discovered in Cretaceous exposures of Southern France (Freytet, 1965; Kerourio, 1981; Cousin et al., 1989, 1994).’

‘Since late 1800s’ should be ‘Since the late 1800s'

Lines 209-211 - ‘thinning by dissolution (Cousin & Breton, 2000). As other titanosaur eggs, the French eggs display nodular rounded structures on the outer surfaces and spherulitic eggshell units with slender…’

‘As other titanosaur’ should be ‘As in other titanosaur'

Lines 289-290 - ‘…Alike the above-mentioned European, Asian, and South American nesting sites, the dense…’

‘Alike the above’ should be ‘like the above'

Lines 340-342 - ‘…substantial increase of surface in contact with the surrounding environment, suggesting they could acted buffering acidic erosion during long periods of incubation, an interpretation consistent…’

‘suggesting they could acted buffering’ should be ‘suggesting they could have acted to buffer'

Line 372 - ‘…These sites are similar on their eggs‟ morphology besides a thinner eggshell thickness in’

‘similar on their’ should be ‘similar in their'

Lines 417-418 - ‘(brush-turkey) nests in wetter environments in coastal Australia regions in respect with its congener Leipoa ocellata Gould, 1840 (malleefowl) that nests in semi-arid settings.’

‘coastal Australia regions’ should be ‘coastal Australian regions'


Lines 434-437 - ‘Rhizoliths and root traces as well as small fossil logs found there (Garrido, 2010a,b; Jackson, Schmitt & Oser, 2013) suggest a floristic abundance that would propitiate ideal conditions for mound-nesting behaviors.’

‘propitiate’ is not the right word here. It means to carry out an activity that is meant to please or appease a deity. Replace it with ‘prmote’ or ‘bring about’.

Lines 438-439 - ‘Modern megapodes construct surprisingly large nests-mound, if compared to their egg and clutch sizes.’

‘nests-mound’ should be ‘nest-mounds'

Lines 455-471 - ‘…ºC, although eggs mostly lie in the range of 32 to 36 ºC (Booth, 1987). In contrast brush-turkey females burrow into the mound vertically. The litter material does not collapse around the tunnel as sand would in a malleefowl mound (D. Booth, pers. comm.). Hence, eggs can be located all over the place in large mounds and not just in the central core as in the malleefowl ones. One strategy would result in eggs grouped together in a compact clutch, while the other results in eggs isolated or lined up in small groups when the mound is eroded away. Interestingly, the latter matches to the pattern already observed by Cousin et al. (1989) and Cousin and Breton (2000), who interpreted that small groups of eggs (2 to 4) were not randomly distributed but belonging to circular “supergroups” of up to 15 eggs and 3.5 m in diameter (see Fig. 10 in Cousin & Breton, 2000) substantially separated from others (Cousin et al., 1989; Cousin & Breton, 2000). Additionally, the few-egg clutches found in close association in Coll de Nargó that were interpreted by Vila et al. (2010) as partially preserved ~25-egg clutches, show similar patterns. As such, the egg spatial distribution, in small clusters linearly to compactly grouped, but contained in round shaped areas of up to 2.3 m (see Table 1; Vila et al., 2010), would either support burrow- and mound-nesting (Cousin & Breton, 2000).

This section talks about the nesting behaviour of extant birds, and then starts to make comparisons with titanosaurs. However, this is confusing because nowhere do the authors tell the reader that they are switching to talk about dinosaurs (I only realise this from the references being cited). The sentence starting ‘Interestingly’ needs to make it clear that the text that follows applies to dinosaurs/titanosaurs.

Also -
‘matches to the pattern’ should be ‘matches the pattern'
‘but belonging to circular’ should be ‘but belong to circular'
‘would either support burrow- and mound-nesting’ - this phrasing does not make sense - where is the ‘or’ to follow the ‘either’?

Lines 498-500 - ‘Interestingly, although Tandon et al. (Tandon et al., 1995) suggested the nesting activity in the Lameta Formation was coeval with the first Deccan lavas, questions regarding its possible relationship still requiring more attention and further research.’

‘Tandon et al. (Tandon et al., 1995)’ should be ‘Tandon et al. (1995)’
‘still requiring’ should be ‘still require'

Lines 530-533 - ‘…alligators, that exhibit similar “in masse” laying behaviors. Moreover, titanosaur hatchlings would likely surface out of their nests synchronously in contrast to one at time as modern megapodes, an anti-predatory strategy that insure species survival.’

‘to one at time’ should be ‘to one at a time'

‘an anti-predatory strategy that insure species survival.’ - do the authors really mean this? This seems to be suggesting that the synchronous emergence of young from the nest is favoured by group- or species-level selection. This would be a contentious claim. While mechanisms for species selection have been discussed in the literature, they are far from being widely accepted. In any case, why interpret this as an adaptation at the group- or species-levels, when there are clear grounds for arguing for selection of this trait at the individual level - any parent that produces young that are more likely to emerge from the nest synchronously is likely to get more of their genes into the next generation.

Some references for an introduction to titanosaurs -

Curry Rogers 2005 in ‘The Sauropods’, eds. K. Curry Rogers and J. Wilson. - A little out of date now, but a good general background.

Wilson, J.A. 2006. In the Salas de los Infantes Dinosaur symposium volume. Provides an overview of titanosaurs.

Mannion P.D. et al. 2011 - Biological Reviews paper. Provides an overview of sauropod diversity through time, and the replacement of non-titanosaurs by titanosaurs in the Cretaceous

Mannion P.D. and Upchurch P. 2010 - paper in Paleobiology entitled ‘A quantitative analysis of environmental associations in sauropod dinosaurs’- Argues that titanosaurs preferred inland habitats to coastal ones.

Wilson J.A. and Carrano M.T. 1999 - paper in Paleobiology - wide gauge stances in titanosaurs.

Mannion P.D. et al. 2013 - paper in Zoological journal of the Linnean Society - phylogeny of Titanosauriform sauropods and review of the origins and spatiotemporal distributions of titanosaurs.

Sander M. et al. 2011. paper in Biological Reviews - Gigantism in sauropods and an excellent Introduction to the ecology of the group.

Benson R.B.J. et al. 2014 - paper in PLos Biology - presents a range of body mass estimates for dinosaurs, including sevral titanosaurs.

Experimental design

See comments on 'Materials and Methods' above.

Validity of the findings

See my general comments under 'Basic reporting'

Additional comments

See my general comments under 'Basic reporting'

---

## Round 0.2 · Minor Revisions

Dear authors,

I have accepted the reviewers decision of 'minor revision'. Please pay close attention to the language recommendations they have made.

·

Basic reporting

See 'Comments to the authors'

Experimental design

See 'Comments to the authors'

Validity of the findings

See 'comments for the author'

Additional comments

This is the second time I have reviewed this MS. The authors have responded positivley to my comments and those of the other reviewer, and I now believe that this paper is ready for publication (subject to some minor revisions). I have found a few typos and grammatical errors (listed below). Once these have been corrected, I think the paper can be accepted.

Specific points

1. Lines 37-38 ‘…they preferred inland rather than coastal habitats. They experienced a great expansion during the Late Cretaceous, chiefly in South America, where they diversified in more than 20…’’

I would replace ‘expansion’ with ‘radiation’.
‘in more than 20’ should be ‘into more than 20’

2. Lines 104-105 ‘need for a full and accurate record prevented include in this study, for example, the findings of isolated eggshells in Morocco and Tanzania, in’

‘prevented include in this’ should be ‘prevented the inclusion in this’

3. Lines 160-162 ‘Although the eggs spatial distribution in two superposed rows, whereas the upper row has a greater amount of eggs (Tauber, 2007; Grellet-Tinner & Fiorelli, 2010; Fiorelli et al., 2012), suggesting that they were deposited in dug out-nests, close to geothermal vents, springs, or pools,’

having ‘although’ and ‘whereas’ in the same sentence is confusing - it is almost a double negative. I recommend rephrasing this for greater clarity.

4. Lines 163-164 ‘…no nesting structure has been preserved (Fiorelli et al., 2013). Therefore, these fossil assemblages are referred only as egg clutches, although the oological data (Table 1) strongly support parental…’

‘referred only’ should be ‘referred to only'

5. Lines 188-189 ‘…sediments are locally tuffaceous (Paik, Kim & Huh, 2012), Fiorelli et al. (2012) suggested that nesting in geothermal settings were not limited to a particular endemic neosauropod population’

‘nesting in geothermal settings were’ should be ‘nesting in geothermal settings was'

6. Line 226 ‘…sauropod (mostly titanosaurs taxa) egg clutches or mixed egg fossils including ornithopod’

‘mostly titanosaurs taxa’ should be ‘mostly titanosaurian taxa'

7. Lines 239-240 ‘partial hatchling found at the exposures near Dholi Dungri locality, Gujarat, India (Mohabey, 1998’

‘near Dholi Dungri locality’ should be ‘near the Dholi Dungri locality'

8. ‘Line 420 ‘Among the titanosaurs nesting sites reviewed in this investigation, only two have been…’

‘titanosaurs nesting’ should be ‘titanosaur nesting'

9. Line 447 ‘…activities. Paradoxically, megapodes’ nest-burrows are dug preferably in soft soils, in’

‘megapodes’ nest-burrows’ should be ‘megapode nest-burrows'

10. Line 460 ‘…minerals are prevalent in continental brines, which could be also formed under high’

‘could be also formed’ should be ‘could also be formed'

11. Line 469 ‘…coastal Australian regions with respect with its congener Leipoa ocellata Gould, 1840’

‘respect with its’ should be ‘respect to its'

12. Line 489 ‘Although crocodilians’ mound-nests also average 1 m high and 3m in diameter’

‘crocodilians’ mound-nests’ should be ‘crocodilian mound-nests'

13. Line 493 ‘…megapodes construct surprisingly large nests-mounds, if compared to their egg and’

‘nests-mounds’ should be ‘nest-mounds'

14. 540 ‘Tandon & Andrews, 2001; Wilson et al., 2010; Prasad & Sahni, 2014), allows to infer an’

‘allows to infer an’ should be ‘allows us to infer an'

·

Basic reporting

The only real problem with the paper as it stands is the English. The other authors have identified most in their reviews; I have identified additional errors as noted in the attached PDF.
I would have liked to have seen a tabulation of megapode taxa, their nesting strategies, and their preferred environment so that they could be compared with the sauropods. However, I am confident that such a table is probably A) available elsewhere, and definitely B) beyond the scope of this paper.

Experimental design

No comments.

Validity of the findings

I think the interpretation is fine. I doubt that any sauropod monitored its nest in quite the way that the modern mallee fowl does, although less fastidious mound building like that of the brush turkey is possible.

Additional comments

No additional comments.

---

## Round 0.3 · accepted · Accept

Dear authors,

Thank you for making the suggested changes by the reviewers. As such, I am happy to recommend your manuscript for publication.